# Ex Vivo Fluorescence Confocal Microscopy (FCM) Ensures Representative Tissue in Prostate Cancer Biobanking: A Feasibility Study

**DOI:** 10.3390/ijms232012103

**Published:** 2022-10-11

**Authors:** Ulf Titze, Johannes Sommerkamp, Clara Stege, Fried Schneider, Christoph Brochhausen, Birte Schulz, Barbara Titze, Furat Abd Ali, Sasa Pokupic, Karl-Dietrich Sievert, Torsten Hansen

**Affiliations:** 1Institute of Pathology, University Hospital OWL of the University of Bielefeld, Campus Lippe, 32756 Detmold, Germany; 2Department of Urology, University Hospital OWL of the University of Bielefeld, Campus Lippe, 32756 Detmold, Germany; 3Institute of Pathology, University of Regensburg, 93053 Regensburg, Germany; 4Central Biobank Regensburg, University Hospital Regensburg, 93053 Regensburg, Germany

**Keywords:** biobanking, prostate cancer (PCa), fluorescence confocal microscopy (FCM), digital medicine, cryoasservation, targeted sampling

## Abstract

Background: Biobanking of prostate carcinoma is particularly challenging due to the actual cancer within the organ often without clear margins. Frozen sections are to date the only way to examine the biobank material for its tumor content. We used ex vivo fluorescence confocal microscopy (FCM) to analyze biobank samples prior to cryoasservation. Methods: 127 punch biopsies were acquired from prostatectomy-specimens from 40 patients. These biopsies were analyzed with a Vivascope 2500-G4 prior to their transfer to the biobank. In difficult cases, larger samples of the prostatectomy specimens were FCM scanned in order to locate tumor foci. After patient acquisition, all samples were taken from the biobank and analyzed. We compared the results of the FCM examinations with the results of conventional histology and measured the DNA content. Results: With upstream FCM, the tumor content of biobank samples could be determined with high confidence. The detection rate of representative biobank samples was increased due to the rapid feedback. The biobank samples were suitable for further molecular analysis. Conclusion: FCM allows for the first time lossless microscopic analysis of biobank samples prior to their cryoasservation and guarantees representative tumor and normal tissue for further molecular analysis.

## 1. Introduction

Prostate carcinoma (PCa) is the most common malignancy and the second leading cause of cancer-related death in men in the United States [1]. From a clinical perspective, prostate cancer represents a highly heterogeneous disease. The clinical courses vary from slow-growing tumors that manifest later in life, remain organ-confined for a long time, and have little impact on life expectancy up to rapidly progressive and fatal metastatic tumors in younger men [2]. Significant profiling studies over the past decade have provided insights into the phenotypic and molecular complexity of primary and metastatic prostate carcinoma [3].

Much of this complexity is rooted in the multi-focal nature of the disease. More than 80% of prostate carcinomas show topographically and morphologically distinct tumor foci [4]. At the molecular level, mutational profiles in primary and metastasized PCa suggested independent origins and different biological behaviors of the tumor foci [5,6]. The goal of future research is to be able to diagnose and treat aggressive prostate carcinomas earlier while sparing men with indolent tumors unnecessary treatments. It is predictable that this will require multi-disciplinary approaches including novel computational techniques and deep learning algorithms to analyze multidimensional data [7].

Biobanks are valuable tools for the careful integration of findings from clinical, imaging, and molecular pathology studies [8]. Corresponding to the growing number of new technologies for research the need for native tissue specimens increases significantly [9,10]. Thus, there is a growing demand for high quality biobank samples from the prostate containing relevant index lesions [5]. Compared to other organ tumors, preservation of fresh tissue from prostatectomy specimens is a major challenge. In many cases, the tumors, which are typically multifocally localized in the relatively small organs, are barely visible macroscopically [11]. For certain research applications, especially cell culture approaches, it is necessary to ensure a high tumor content in native samples. For quality control, the preparation of frozen sections is currently the gold standard, which places high burdens on the technical equipment and qualification of the personnel which in turn may lead to significant tissue losses.

Ex vivo fluorescence confocal microscopy (FCM) is a novel digital microimaging technique that provides optical sections of unfixed tissues [12]. The images closely resemble frozen sections and allow timely and lossless histological examination of surgical specimens and biopsies. The technique is well established in dermatology for the histological diagnosis of skin tumors and inflammatory dermatoses and also shows promising results in other fields of histopathology [13,14,15] and especially in the diagnosis of prostate carcinoma [16,17,18].

This prospective study presents our practical experience with the use of FCM and its significant value in the preparation of biobank specimens from prostatectomy specimens. The primary endpoint of our investigation was concordance in the assessment of biobank specimens in FCM and conventional histology. The secondary endpoint was to determine whether higher rates of representative biobank specimens could be obtained by targeted random testing of prostatectomy specimens.

## 2. Results

### 2.1. Available MRI Data

MRI imaging was available in only 20/40 (50%) patients. The images showed between one and three morphologically delimitable foci (11 patients with 1 focus, 8 patients with 2 foci, and 1 patient with 3 foci). One patient had a single PIRADS 3 lesion (P23). Findings from 12/20 patients were assigned to PIRADS group 4. Moreover, 7/20 patients had PIRADS group 5 findings.

### 2.2. Tumor Extension in the Prostatectomy Preparations

The specimens from 24/40 patients showed locally limited tumor disease (1× pT2a, 3× pT2b, 20× pT2c). Locally advanced tumor disease was present in 16/40 patients (10× pT3a, 6× pT3b). The localized tumor diseases had a mean tumor volume of 4.3 ± 3.4 mL (degree of infiltration 10 ± 7.3%) and mostly showed multi-focal foci (80% with ≥1 tumor foci, two patients with 6 foci). Locally advanced tumors showed larger tumor volumes (11.3 ± 11.1 mL in stage pT3a, 24.2 ± 19.8 mL in stage pT3b) and higher degrees of infiltration (23.5 ± 15.7% in stage pT3a, 43.0 ± 26% in stage pT3b) with a higher tumor stage. Here the tumor manifestations showed a tendency towards confluence of the foci (50% of cases with ≥2 tumor foci in stage pT3a, 33% of cases with ≥2 tumor foci in stage pT3b).

### 2.3. Identification of Tumor Foci

Tumor foci could be identified on the slices of the prostatectomy specimen in 36/40 (90%) patients by integration of all available information from prostate biopsies, MRI imaging (if available), and macroscopic findings including palpation und upstream FCM analysis. Representative tumor and normal tissue were obtained more frequently in the cohort of patients with available MRI images than in the cohort without an MRI dataset (19/20 (95%) vs 17/20 (85%)).

In the cohort of patients with MRI images, the tumor was already detected in the biopsies from the PIRADS-classified foci in 17/20 (85%) patients. Punch biopsies could be obtained specifically from the tumor foci and normal tissue. After confirming the content of tumor glands and normal tissue with FCM, the samples were stored in the biobank. In three of these patients, the tumor was not detectable by this method. FCM analysis of the punch biopsies taken from the MRI foci showed exclusively tumor-free parenchyma with stromal nodules. In these cases, individual quadrants were scanned based on the result of the biopsies and macroscopic examination. This made it possible to identify and subsequently biopsy the tumor in two patients (P01, P22). One patient (P23) showed only small multi-focal tumors with a total volume of 0.6 mL in the prostatectomy specimen, which could not be found by the additional examinations.

In the cohort of patients without MRI images, tumor foci could be identified in 15/20 (75%) cases based on macroscopic examination/palpation and taking the biopsy result into account. The presence of representative tumor prior to transfer to the biobank was confirmed in the FCM of the punch biopsies. In the other five of these patients, selected quadrants were examined with the FCM. The built-in camera provided macroscopic images, in which the parenchyma is visualized with high magnification. Compared to the unaided eye, the images clearly distinguish conspicuously condensed tissue sections from the fibrous structure of the surrounding tissue. These conspicuous sections can then be specifically analyzed at the microscopic level (Figure 1). Carcinoma foci could be found this way in 3/5 cases. In two cases, no tumor was detected even with this approach. In one patient (P18) unifocal tumor, the search was performed on the wrong site as the localizations of the tumor-involved biopsies were incorrectly taken from the correct biopsy report (sampling error). Another patient (P20) showed a highly differentiated carcinoma, which was mistaken for stromal hyperplasia in the FCM scans due to insufficient image quality.

### 2.4. FCM Analysis of the Biobank Samples

A total of 127 punch biopsies were collected from the prostatectomy specimens of the 40 patients (mean 3.1 ± 1.0 samples, range 2–6 samples). One hundred and twenty-six punch biopsies were analyzed by FCM before cryoasservation. One biopsy (P26 LC2) was not scanned due to time constraints, as it was undoubtedly from a detected tumor site. Digital images showed tumor foci in 58 of the specimens and 68 specimens showed tumor-free glandular parenchyma on FCM. Moreover, 43/58 (74%) of the biopsies showed tumor in >40% (full hits), and 15/58 (16%) biopsies revealed tumor in <40% (partial hits) of the patients. 

We diagnosed representative tumor in the scans of at least one sample in 37/40 (93%) patients. All obtained punch biopsies of these patients (tumor and normal tissue) were snap frozen and preserved in the biobank. In 3/40 (7%) patients (P18, P20, and P23), no tumor manifestations could be documented in the FCM scans. The samples from these patients were not cryopreserved in the biobank but were processed together with the prostatectomy specimen according to the FFPE procedure.

### 2.5. Processing Times for FCM-Analyses

The duration of sample preparation was about 2 min for small punch biopsies sizes (10 × 4 × 4 mm). The scanning process together was another 2 min, so we came to a total duration of 4–5 min per scan. FCM examination of entire quadrants (25 × 20 × 5 mm) resulted in slightly longer preparation times of about 3 min, as it is more time consuming to align the large materials evenly on the glass slide. The scanning process of such large objects took between 6 and 8 min. In total, it took about 10–11 min to scan an entire quadrant.

The total duration of FCM examinations varied greatly from case to case (Appendix A, Appendix A). In this feasibility study, several sample preparation strategies were tested to reduce the overall duration of the FCM analyses. In particular, several options were explored to analyze punch biopsies from different collection sites together on one slide. In some cases, tumor and normal tissue could be obtained with just four punch biopsies. These could then be analyzed within 5 min in a joint scan. In difficult cases, e.g., large organs with hyperplastic stromal nodules, up to six scans were made, each with several punch biopsies or whole quadrants, resulting in processing times of up to 44 min (mean 15.6 ± 10.6 min). The total time to cryoasservation of the biobank specimens after receipt of the prostatectomy specimen varied greatly and was also dependent on the frozen section specimens requested. The mean ischemia time was 49.5 ± 21.3 (range: 20–109) minutes.

### 2.6. Histological Quality Control of the Biobank Material and Comparison with the Prostatectomy Specimens

The duration of storage in the biobank ranged from 96 days for P40 to 322 days for P01. All biobank samples of tumor and normal tissue were completely FFPE processed. No further material remained in the freezer. After DNA extraction from one tumor and one normal tissue sample, the residual material and the remaining cryo-assayed biobank samples from all patients were processed according to the FFPE procedure and examined histologically. The histological sections of the punch biopsies were compared with the corresponding FCM scans (Figure 2).

On statistical analysis (Table 1), matching diagnoses regarding tumor presence were found in 96.8% of the biobank specimens (kappa = 0.93). False positive FCM assessments occurred in patient P03; here, regenerative gland proliferates were incorrectly assessed as well-differentiated tumor in FCM analysis. Conversely, no tumor tissue was represented in the residual material of a biobank sample from patient P22 after DNA extraction. False-negative FCM assessments resulted in two patients. In patient P07, microfocal incipient carcinoma foci (<5%) were detectable in a sample from normal tissue in the final FFPE evaluation. However, other tumor-free samples from normal tissue were available in this patient. In patient P20, well-differentiated tumor glands in one of the specimens were incorrectly interpreted as hyperplasia due to insufficient image quality of the scan. This led to the fact that in this patient no biobank material was cryoasserved.

The tumor-infiltrated punch biopsies showed a mean infiltration grade of 60.5 ± 33.2% (range 1–100%) in histological analysis. Moreover, 42/59 (71%) tumor-infiltrated punch biopsies had > 40% tumor (full hits), 17/59 samples were partial hits (<40% tumor content). For the estimation of tumor content in FCM, there were matches in 85% of samples (kappa = 0.63). In 5/59 (8%) punch biopsies, tumor content was overestimated in the FCM images, and in 3/59 (5%) samples, it was underestimated.

In the overall analysis, 90.5% of biobank samples could be correctly classified (kappa = 0.84) using FCM with regard to the presence of tumor and tumor content. In this way, representative samples of normal and tumor tissue were specifically preserved from 36/40 (90%) specimens. In three of these cases, cryomaterial preservation was omitted due to a lack of tumor evidence in FCM. One of these cases (P23) had localized tumors with comparatively small tumor volume (0.5 mL) in the prostatectomy specimens. A sampling error occurred in one case (P23, tumor volume 9.1 mL). The third case (P20, tumor volume 2.4 mL) was discarded due to a false-negative FCM assessment. In one case (P03), a false-positive FCM diagnosis resulted in the preservation of exclusively tumor-free parenchyma.

### 2.7. DNA Content of Biobank Samples

For each patient with representative biobank samples, DNA was extracted from one tumor and one normal tissue sample and analyzed for quantity and purity (Appendix A, Appendix A). Sufficient amounts of DNA were obtained from all tissues for further molecular analyses (293.4 ± 165.9 ng/µL, range: 12.6–590 ng/µL). The UV spectra as well as the A260/280 and A260/230 ratios indicated appropriate purity for subsequent molecular analyses. 

## 3. Discussion

Biobanking of fresh tissue is one of the most important bases for translational research approaches as well as recent primary cell culture techniques. Native tissue processing is a time-consuming process that places high demands on technical equipment and requires the use of highly skilled personnel. In addition, compared to other organ tumors, prostate carcinoma poses a particular challenge for biobanking. Prostate carcinomas are often difficult to recognize macroscopically, in the majority of cases multifocally localized in the usually only chestnut-sized prostatectomy specimens and may additionally be superimposed with hyperplastic stromal nodules, especially in enlarged organs [11].

Numerous methods have been proposed for tissue sampling from prostatectomy specimens [19,20,21,22]. These models were based on random tissue sampling and/or visual and palpatory examination primarily from the peripheral zones of the prostate, where carcinomas are localized in the vast majority of cases. For sample collection for the International Cancer Genome Consortium (ICGC), every second slice of the prostatectomy specimen was completely preserved [22]. This resulted in a high number of samples with detected tumor, but only 15% of the samples contained a tumor content of >10%. A synopsis and computerized analysis of the methods available so far in 2017 highlighted the need to reduce the number of biobank samples collected and suggested the integration of imaging and biopsy data [23]. By correlation with MRI imaging, tumor foci in prostatectomy specimens could be localized and targeted for biopsy, resulting in more effective specimen collection with up to 70% of suitable samples while sparing surgical margins [24].

We clearly demonstrated higher detection rates of prostate cancer in patients who had preoperative MRI imaging in our relatively small cohorts. The yield could be further increased by FCM analysis resulting in presentative samples of both normal tissue and tumor in 90% of the patients. It was recognizable already while processing whether representative material was obtained or whether a second attempt was necessary. In difficult cases, the Vivascope 2500-G4 helped to locate tumor foci in the prostatectomy specimens. The built-in digital camera provided high-quality and highly detailed macroscopic images of the parenchyma. Firm parts were distinguishable from surrounding fiber-rich tissue and could be subsequently examined at the microscopic level. In favorable cases, tumors could be targeted for biopsy in correlation with the macroscopic image. In two of three cases in which no tumor was initially detectable, representative biobank samples were obtained by this second attempt. The additional processing times ranging from 5 min in clear cases to 44 min in difficult specimens seemed tolerable in regard to the prolonged ischemia time and its impact on RNA integrity [21].

In the protocols established so far, the quality control of the biobank samples usually took place after cryofixation of the tissue. Often, samples were not checked for suitability until the enrollment phase of scientific projects. Better quality control was accomplished by the use of punch biopsy techniques, in which the removal zones were identifiable in the prostatectomy specimens and allowed orienting conclusions to be drawn about the acquired tissue in the biobanks [25]. Regardless, all specimens should be quality controlled prior to their use for research. This is usually achieved in frozen sections, but again this results in losses of tissue.

Ex vivo confocal microscopy provided a solution to this dilemma. As demonstrated in previous studies, the presence of PCa as well as the tumor content could be determined with a high degree of certainty using the digital scans [26,27,28,29]. This system offered pathologists a completely new perspective in the practice of biobanking as—for the first time—biobank samples could be examined microscopically without a loss prior to their cryo-conservation. The FCM images were used to assess whether the sample required further processing, e.g., macrodissection, prior to inclusion in scientific studies (“What you see is what you get!”). After image acquisition, the tissue was available as native material and could be cryofixed on average within 15 min (maximum 44 min) after collection.

A possible caveat to the method arises from the use of AO as a nuclear stain. The slightly acidic dye readily penetrates cell membranes and accumulates in acidic environments. In living cells with preserved pH compartmentalization of cell organelles, AO labels lysosomes and intercalates with nucleic acids. In contrast, fixed, apoptotic, and malignant cells show increased cytoplasmic AO accumulation. This is used diagnostically for fluorescence-based intraoperative tumor detection and results in novel approaches for photo- and radio-dynamic therapies [30]. Although mutagenic effects have been described in bacteria, recent pilot studies have confirmed its safety in clinical use [31,32]. According to the International Agency for Research on Cancer, there are currently (as of 1998) no adequate data or evaluations on the carcinogenicity of AO [33]. Limited in vitro studies exist in which the non-interference of subsequent RT-PCR analyses by AO has been documented [34]. In cell cultures, the vital dye has been used as an indicator of the state of acidic vesicular organelles in autophagy studies [35]. AO is cytotoxic only at high concentrations (1.0–10 mM) [36]. Since data on mutagenicity remain unclear at this time, comparative studies on tissue material with and without AO pretreatment are necessary to rule out confounding influences on sequencing studies. Our previous study showed a penetration depth of 100 mm with an incubation time of 30 s [37]. Thus, for a biopsy cylinder of 4 mm diameter and 6 mm length, approximately 87.2% of the tissue (9.6 mm³ stained volume, 65.8 mm³ unstained volume) remains unstained. According to the manufacturer, subsequent incubation of the tissue with ethanol also appears to be a promising means of eliminating AO from the tissue—systematic studies should also be carried out in this regard.

In conclusion, FCM analysis allowed high efficiency in obtaining biobank samples. The upstream microscopic control guaranteed that representative tumor and normal tissue could be obtained from prostatectomy specimens in a small number of samples (mean 3.1 ± 1.0). It can be deduced that similar results can also be achieved in other demanding organ systems, such as in the biobanking of breast carcinomas.

A further advantage is given by the fact, that the images are immediately available in digital form and can be archived together with the rest of the clinical and morphological data. The combination of clinical data, image data from MRI imaging, native biobank material, and directly associated microscopic findings are a promising basis for the construction of artificial intelligence-based approaches [38,39].

## 4. Materials and Methods

### 4.1. Patients

Forty male patients aged 66 ± 6.6 years (range 53–78 years) participated in this study. The mean PSA values were 11.7 ± 11.8 ng/mL (range 2–59 ng/mL). Prostate carcinoma was confirmed in prostate biopsies in all patients and indications for prostatectomy had been established in interdisciplinary tumor conferences. All participating patients were informed about the examinations in the study and provided signed informed consent. During the informed consent process, it was emphasized that all cryoasserved tissue samples were processed for retrospective analysis after 6 months at the latest and that no material remained in the biobank for later investigations. All investigations in the study were performed in accordance with the ethical principles of the WMA Declaration of Helsinki.

### 4.2. Study Design

As part of the macroscopic processing of prostatectomy specimens for routine histology, regions of tumor and normal tissue were identified for this study, and targeted punch biopsies were obtained from these areas for the biobank (Appendix A, Appendix A). The obtained specimens were preliminarily analyzed by FCM to determine the content of tumor or normal tissue. Where representative tumor and normal tissue could be obtained in each case, the materials were immediately transferred to the biobank. In cases where representative tumor tissue could not be identified, cryoasservation was not performed and samples were FFPE processed along with the prostatectomy specimen.

Three months after the last subject was enrolled, the preserved tissues were retrieved from the biobank and analyzed. For each sample, tumor content was determined histologically in sections of the formalin-fixed paraffin-embedded (FFPE) material and statistically compared with the result of intraoperative FCM analysis. For additional quality control, the collection sites were identified in the prostatectomy specimens and matched to the punch biopsy specimens. We additionally determined the DNA content and purity of representative samples from tumor and normal tissues as a measure of the quality of the biobank samples.

### 4.3. Processing of the Prostatectomy Specimens

The prostatectomy specimens were transferred immediately as fresh tissue to the Institute of Pathology of the University Hospital OWL of the University of Bielefeld. Surgical pathology dissection of all specimens as well as FCM examinations were performed by a senior pathologist (U.T.). The organs were measured in three dimensions and weighed. The prostate volumes were measured using Archimedes’ principle. After the side-separated ink marking of the surgical margins, tangential sections were obtained from the prostatic apex and base for intraoperative frozen sections. After separation of the seminal vesicles and ducts, 4/5 of the prostate was serially sectioned in the transverse plane from apex to base in 4–6 mm intervals [40,41]. Each slice was divided into four systematically designated quadrants (L (left)/R (right)—slice A-F—1 (anterior)/2 (posterior)). The prostate base and parts of the seminal vesicles were completely dissected in parasagittal sectioning. After the punch biopsies (biobank specimens) were collected, the slices were placed in prepared cassettes and processed according to standardized FFPE procedures (Figure 3). Histological evaluation was performed promptly by a specialized pathologist (TH) according to current guidelines.

### 4.4. Obtaining Biobank Samples

One to three punch biopsy cylinders were collected from the identified target areas using a 4 mm punch biopsy needle (Stiefel Disposable biopsy punch Ø = 4 mm, GSK Consumer Healthcare, Ireland) [42]. The samples were analyzed for tumor content using FCM. Once representative material of tumor and normal tissue was obtained, the specimens were placed on cork plates without mounting medium and cryo-fixed with precooled isopentane in a cryostat (−18 °C). After completion of the routine procedures, the frozen biobank samples together with the cork plates were transferred to suitable sample tubes and stored at −80 °C for the long term. 

### 4.5. Ex Vivo Confocal Microscopy

The VivaScope 2500-G4 (Vivascope GmbH, Munich, Germany) microscope was used for FCM examinations of the biobank samples. This confocal laser scanning microscope was designed for intraoperative examination of unfixed tissues. The native materials require a short pretreatment with a fluorescent dye (AcridinOrange, AO; 0.6 mM; Sigma-Aldrich^®^, St. Louis, MO, USA). The same staining protocol was used as in previous publications (1. surface protein denaturation with ~10 s. Ethanol; 2. nuclear staining with AO for ~30 s, 3. removal of excess dye in saline solution for ~10 s [14]). The tissue is not damaged by the procedure and is available for all subsequent histological, immunohistological, and molecular studies.

AO is a fluorochrome with DNA-intercalating characteristics and is used as a vital dye for the examination of living cells and tissues [30]. The DNA dye complexes have an excitation maximum at 470 nm and an emission maximum at 526 nm [43]. AO shows unique metachromatic properties with differing fluorescence signals when bound to RNA (orange) and DNA (green).

Illumination is performed with two lasers illuminated at defined wavelengths (488 nm and 638 nm). By detecting the fluorescence signals, the cell nuclei are visualized. With the long wavelength laser, cytoplasmic and extracellular are displayed in reflection mode. Both the reflectance and fluorescence signals are detected and combined simultaneously. A built-in algorithm translates the signals into a pseudo-colored image in which cell nuclei are rendered blue and cytoplasmic/extracellular structures are rendered reddish [44]. The resulting image is very similar to hematoxylin-eosin-stained cryostat sections.

The microscope is equipped with a 38× water immersion objective with a numerical aperture of 0.85. Total magnifications of up to 550× are achieved. According to the manufacturer, specimens up to 2.5 cm × 2.5 cm in size can be examined. In addition, the VivaScope 2500-G4 is equipped with a digital camera that provides macroscopic images of the tissue samples prior to microscopic examination. The macroscopic images correlate exactly with the microscopic images and allow navigation and selection of the area to be examined.

### 4.6. Identification of Tumor Areas

Tumor areas were identified in combination with the results of previous biopsies and macroscopic examination or palpation. When MRI data were available, images were correlated with slices of the prostatectomy specimen as described [24], and punch biopsies were taken from PIRADS-classified lesions.

In cases where representative tumor areas could not be identified in this manner, selected quadrants were fully examined with FCM. The macroscopic images allowed a very detailed assessment of the tissue structure. Parts of the quadrants containing tumor-suspected structures could be examined microscopically. The identified tumor areas of this quadrant could be subsequently targeted for biopsy by correlation with the macroscopic image.

### 4.7. Examinations of Prostatectomy Specimens

The prostatectomy specimens were promptly examined as part of routine practice and evaluated according to the recent guidelines. The tumor foci were additionally visualized in schematized maps of the prostate [45]. This visualization was used to determine the number of morphologically distinguishable tumor foci. Furthermore, for each slice, we calculated the percentage of the tumor in the total area and concluded the percentage of the tumor in the total volume. For given prostate volumes, tumor volumes for each patient were expressed according to the percentage [46].

### 4.8. Assessment of Tumor Content in the Biobank Samples

All biobank specimens were fixed in buffered formalin and histologically processed according to a standardized FFPE protocol. The hematoxylin-eosin-(HE)-stained slides were carefully assessed by two experienced pathology specialists (B.S., Ch.B.). The presence of tumor (yes/no), ISUP-grade, and the tumor content in 5% increments were documented. Additionally, all FCM images were re-evaluated by a second pathologist (B.T.) and the presence of tumor, ISUP-grade, as well as tumor content were documented accordingly.

### 4.9. Statistical Analysis

The presence of tumor in the biobank samples was evaluated in a binary variable system (0: no tumor; 1: evidence of tumor). The tumor content was transferred to an ordinal scale in view of the suitability of the biobank samples: if the tumor content was >40%, the sample was assumed to be unrestrictedly suitable (“full hit”). Samples that had less than 40% tumor content were considered ‘partial hits’ and could potentially be used for sequencing following macro-dissection. For statistical analysis, biobank samples were classified into 3 categories (0: no tumor; 1: 1–40% tumor; 2: >40% tumor). The levels of agreement between FCM analyses and FFPE diagnoses were measured with Cohen’s kappa [47] and assessed using Landis and Koch’s categories [48].

### 4.10. Assessment of DNA Content in Biobank Samples

A representative sample from the tumor and normal tissue was selected from each patient. Twenty to forty frozen sections (thickness 10 µm) were made from the samples prior to the final histological processing. The remaining material was subsequently processed according to standardized FFPE procedures (Appendix A, Appendix A). 

The sections were mixed in 30 µL proteinase K and 300 µL lysis buffer (Promega) and incubated for 30 min at 56 °C. Extraction was performed using the Maxwell^®^ RSC Blood DNA Kit (AS1400; Promega) in the Maxwell^®^ RSC 48 extraction instrument (Promega) according to the manufacturer’s instructions and using 50 µL of elution buffer. The Qubit dsDNA HS Assay Kit (ThermoFisher Scientific) was used for fluorometric concentration determination of extracted DNA using the Qubit 4 Fluorometer (ThermoFisher Scientific). To assess sample purity, an absorbance spectrum from 200 to 400 nm was recorded on the NanoPhotometer^®^ NP80 (Implen) and the A260/A280 and A260/A230 ratios were calculated. The A260/280 ratio is a good indicator of protein contamination. A value of ≥1.8 indicates that the sample contains pure nucleic acids. An A260/230 ratio of less than 1.8 indicates that the contamination is likely due to organic compounds or chaotropic substances that absorb at 230 nm.

## 5. Conclusions

Our finding revealed the Vivascope 2500-G4 as a valuable tool for cancer biobanking. The combined macroscopic and microscopic examination of selected parts of surgical specimens can help to detect and selectively biopsy difficult-to-identify tumor foci and increase the yield of representative biobank samples. In summary, more than 90% of the biobank samples could be correctly classified with regard to the presence of tumor and tumor content by FCM. After FCM examination, the punch biopsies were available as native material that could be transferred to the biobank without loss. Sufficient amounts of DNA with appropriate purity could be obtained from all of the preserved specimens for subsequent molecular analyses.

The key advantage over the standard procedure is the ability for microscopic examinations of the biobank samples prior to the cryoasservation. The tumor content can be determined with a high degree of certainty. This offers the possibility to decide in advance which materials should be transferred to the biobank. In contrast to frozen section examinations, FCM is a relatively simple and rapid procedure that preserves the examined samples as native material without loss and guarantees high-quality biobank samples with representative tumor and normal tissue. The high-resolution microscopic images of the biobank samples are immediately available and can be archived together with the tissue.

## Figures and Tables

**Figure 1 ijms-23-12103-f001:**
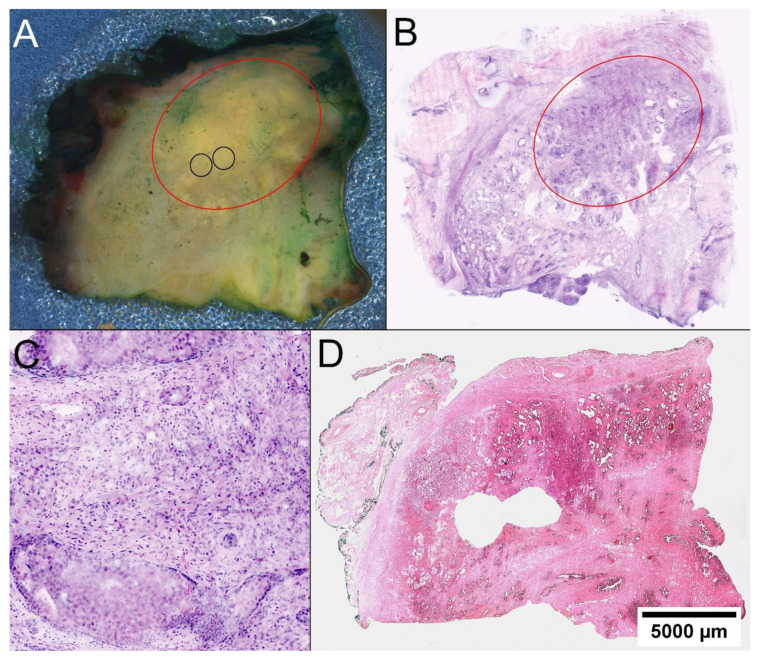
Locating PCa in prostatectomy specimens. (**A**) On the macroscopic image of the quadrate, a discontinuous, yellowish imposing solid foci is distinguishable from the whiter appearing fibrous stroma red circle. Knowing the FCM scan, the tumor was targeted for biopsy (black circles). (**B**) In the FCM scan of the entire quadrant, a well-circumscribed dense tumor tissue demarcates from the surrounding glandular parenchyma. (**C**) At higher magnification, carcinoma infiltrates reveal poorly formed glands and an intraductal tumor component. (**D**) The histological section of the quadrant confirms the sampling of representative tissue from the tumor.

**Figure 2 ijms-23-12103-f002:**
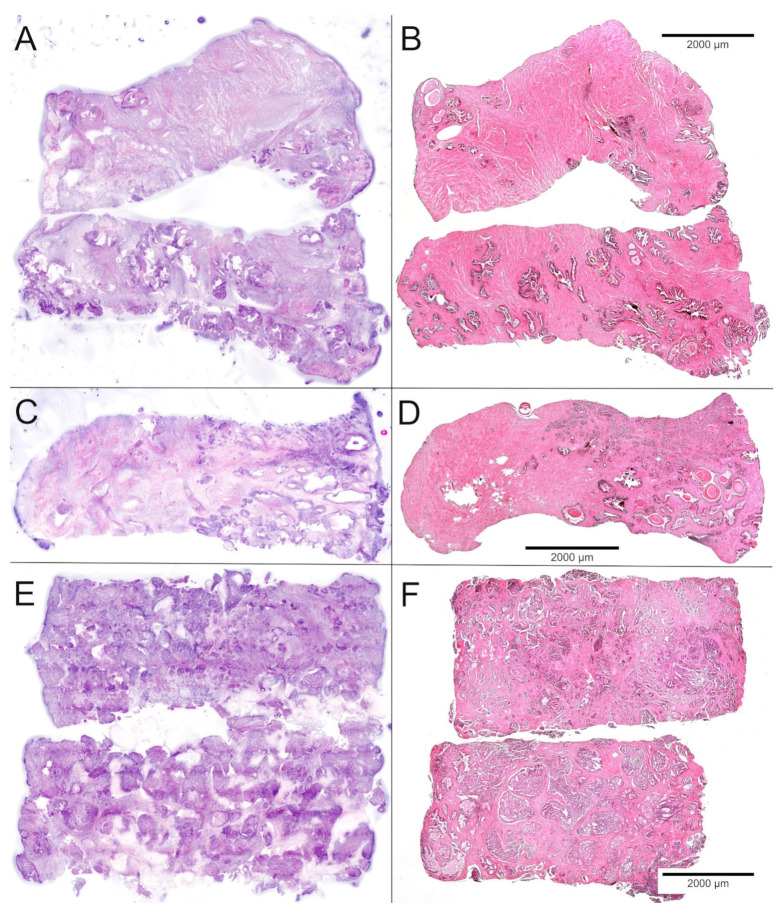
Comparison of FCM scans (**left**) and FFPE sections (**right**) of the biobank samples. (**A**,**B**) Samples of normal tissue showed stromal nodules and regularly shaped glands with luminal corpora amylacea. (**C**,**D**) Partial hit with grouped infiltrates of ISUP-grade 1 carcinoma in 25% of the sample. (**E**,**F**) Full hit revealing 90–100% of poorly differentiated carcinoma in two punch biopsies.

**Figure 3 ijms-23-12103-f003:**
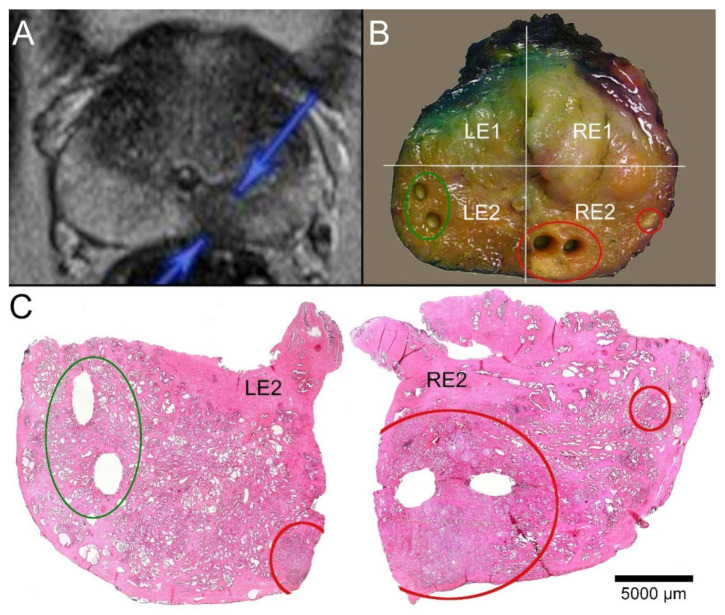
Correlation of MRI imaging, macroscopy, and microscopy. (**A**) T2-weighted MRI of a 59-year-old patient (P32). A round, echo poor PIRADS4 lesion of 11 mm in diameter is marked on the right dorsal peripheral zone (blue arrows). (**B**) Macroscopic dissection of the prostatectomy specimen revealed a slightly brighter lesion in a slice close to the base. A smaller similar focus was visible more laterally. The slice was divided into quadrants (right ventral: RE1, right dorsal: RE2, left ventral LE1, left dorsal LE2). The tumor-suspect foci were localized in the quadrant RE2 (red circles), and the larger lesion was biopsied twice. Another two biopsies representing normal tissue from the contralateral quadrant (green circle) were acquired. (**C**) The paraffin sections of the quadrants LE2 and RE2 showed the sampling localizations of the biobank materials without any doubt. The suspicious foci in RE2 revealed carcinoma infiltrates. The larger tumor focus was represented by the biopsies. The biopsies from the opposite side contained parenchyma free of tumor.

**Table 1 ijms-23-12103-t001:** Comparison of the FCM ratings and the histological examination of the biobank samples.

			FCM-Ratings		
			0	1	2
**FFPE-** **ratings**	68	**0**	66	1	1
	58	**1**	1	11	5
		**2**	1	3	37
	n = 126		68	15	43
Tumor detection (0 vs. 1; 2)			96.8%	ĸ = 0.94	(Nearly perfect)
Tumor content (1 vs. 2)			85.7%	ĸ = 0.64	(Substantial)
Overall rating (0 vs. 1 vs. 2)			90.5%	ĸ = 0.84	(Nearly perfect)

Both FCM images and FFPE-slides were available for n = 126 biobank samples. The obtained tissue was examined with view to the presence and the content of tumor and classified on a three-part ordinal scale (0: no tumor; 1: 1–40% tumor; 2: >40% tumor). The ratings of the FCM and the FFPE material were statistically compared using Cohens’ Kappa.

## Data Availability

Not applicable.

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
