# Peer review of "Ex Vivo Fluorescence Confocal Microscopy (FCM) Ensures Representative Tissue in Prostate Cancer Biobanking: A Feasibility Study"

_ijms, 2022, doi:10.3390/ijms232012103_

Round 1

Reviewer 1 Report

Strengths of the manuscript

The strengths of the manuscript are more important than the suggestions for improvement.

- P10 L320-323: The overall results of this study: 90.5% of biobanked samples were correctly classified (tumor vs. normal) by fluorescence confocal microscopy. Classified samples were cryopreserved in 36/40 prostatectomy specimens (90%).  This is much higher than results obtained in the literature. 

-  This manuscript shows how much improvement MRI can contribute to the task of localizing tumor on gross (macroscopic) fresh tissues examination. As the authors mentioned, it is a not very well- known advantage of this method.

==========================================================

Weaknesses and suggestions for improvement of the manuscript

- The section 2.2 Study design (P2 L88-104) does not include a Figure to illustrate the workflow used in the study with the time/duration of every step of the protocol.

- The section 2.2 Study design (P2 L88-96) does not specify how tumor and normal foci were identified. Details are provided only for tumor on P5 L165-175. Size and number of tissue cores were described on P4 L133-135.

Questions:

- Are tumor punch biopsies representative of the index tumor?

- How long after serial sectioning of the fresh prostatectomy specimens the targeted punch biopsies were obtained?

==========================================================

- The section 2.2 Study design (P3 L97-104) is not very clear about the processing of biobanked punch biopsies after three months of the last subject enrollment.  

Questions:

- Were both biobanked punch biopsies (tumor and normal) entirely submitted to FFPE for histological evaluation and further comparison with FCM findings? Or just the biobanked tumor samples?

- Were tumor punch biopsies representative of the index tumor?

- How long after serial sectioning of the fresh prostatectomy specimens the targeted punch biopsies were obtained?

==========================================================

- The section 2.4 Obtaining biobank samples (P4 L33-140) is not very clear about what happened to the stored biobanked punch biopsies. It states that samples were stored at -80ËšC for the long term after completion of routine procedures.

Questions:

- How long after biobanking the frozen biopsies were retrieved from storage?

-Were all tumor and normal biobanked biopsies entirely submitted to FFPE for histological evaluation after 3 months of last subject enrollment?

==========================================================

- Additional information that may be helpful to include under the section 2.5 Ex Vivo Confocal Microscopy (P4 L141-164).

- Manufacturer of the VivaScope 2500-G4 microscope utilized in the study.

- Source/Product No. of Acridin Orange (AO) dye.

- Specify in seconds AO pretreatment time of targeted samples. P5 L144 described pretreatment with AO as Ë‚1 minute.

- Is there any washing step after AO staining?

- Does AO staining interfere with further establishing cell lines from the targeted samples prior to biobanking?

==========================================================

- The section 2.7 Examinations of prostatectomy specimens states on (P5 L178-179) “…The tumor manifestations were additionally visualized… “

- Instead of “tumor manifestations,” “tumor foci” may be more appropriate.

==========================================================

- The section 2.10 Assessment of DNA content in biobank samples states on (P6 L202-205) “A representative sample from the tumor and normal tissue was selected from each patient. 20-40 frozen sections (thickness 10µm) were made from the samples prior to the final histological processing.” Detailed information about number of biopsy cores per tumor and per normal appears until section 3.6 DNA content of biobank samples on P11 L329-331.

Questions:

- Clarify that DNA was assessed in 1 out of 3 unfixed, fresh, collected biopsies of tumor and normal. Details regarding the number of biopsy cores per region of interest are mentioned until section 3.6 DNA content of biobank samples on P11 L329-331.

- Was DNA assessed in biopsies prior to AO staining?

- Was DNA assessed in biopsies prior to biobanking?

==========================================================

- Regarding title of section 3.3 Identification of tumor manifestations, included on P6 L233:

- Instead of “tumor manifestations,” “tumor foci” may be more appropriate.

========================================================================

P12 L389 of section 4 Discussion states “…The presence of tumor manifestations as well as the tumor content could be determined with a high degree of certainty using the digital scans…’

- Instead of “tumor manifestations,” “tumor foci” may be more appropriate.

==========================================================

P13 L392-394 of section 4 Discussion states “After image acquisition, the tissue was available as native material without loss and could be cryo-fixed within half an hour after collection”.

- Were all cases submitted to biobanking within 30 minutes of the collection of the targeted biopsies?

- Was the ischemia time of each case recorded from the arrival of fresh prostatectomy specimen, including the prostate slicing, the AO staining, and the FCM scanning for image acquisition prior to biobanking?

- How long (in minutes) does the entire procedure take from beginning to end?

==========================================================

Additional comments:
- 6 mm cores might be too big a diameter for cores, risking eliminating a tumor for regular histology. Did you mean 6 mm  length of the core?

The method may not lend itself to use for collection of patient derived xenografts due to acridine orange toxicity: Lin, YC., Lin, JF., Tsai, TF. et al. Acridine orange exhibits photodamage in human bladder cancer cells under blue light exposure. Sci Rep 7, 14103 (2017). https://doi.org/10.1038/s41598-017-13904-0

- More details may be necessary in the the second paragraph starting on line 170

Somewhere in the manuscript, before Table 2 the authors could explain for the general readership what is the meaning of A260/280 reading below 1.8. The same goes for A260/230 ratios.

- Some words like "quadrate" instead of "quadrant" should be checked for terms used by pathologists in the gross room.

Reviewer 2 Report

In their manuscript entitled “Ex Vivo Fluorescence Confocal Microscopy (FCM) ensures rep-2 resentative tissue in Prostate Cancer Biobanking: A feasibility 3 study” Titze and colleagues evaluate the usefulness of ex vivo Fluorescence Confocal Mi-20 croscopy (FCM), i.e. Vivascope imaging, to detect prostate cancer prior to biobanking of the tissue.

The manuscript is well written and structured. The experiments / data acquisition are well planned and conducted. It presents interesting data of another potential area of application of the Vivascope technology, which has demonstrated usability in different setting, including evaluation of biopsies.

I do not have major points, only some minor points that should be addressed:

1.       Why is the threshold of calling a specimen a “full hit” at 40%? This is a rather unusual threshold.

2.       The manuscript can be shortened throughout, as some passages are redundant in parts. This also applies for some data presentation.

3.       In line 320 Kappa value is given as but in table 1 it is 0.8a. Please recheck all numbers.

4.       I think Table 2 can be moved to Suppl. Material.

5.       The Part 3.7 should not be part of the Results’ section but rather after the discussion section.

6.       I am missing information about scan time and length of time of the total Vivascope process in all cases. Please add these data.

7.       Please also critically discuss the extra time needed for Vivascope evaluation, which can be profound in large specimen and thus hinder implementation in routine frozen section.

Reviewer 3 Report

The actors describe the use of FCM to assist diagnostic clinical management of PCa Biobank samples and correlate their observations with FFPE histology.

Overall the study is sound with few minor corrections to the text outlined below. The claims of novelty of their observations is a little overstated as there are other studies that have the used of the vaScope 2500-G4 for prostate tumour identification ex vivo. In fact the same author/s has a similar publication demonstrating similar results, albeit with a different emphasis; Titze, U.; et al,  Diagnostic Performance of Ex Vivo Fluorescence Confocal Microscopy in the Assessment of Diagnostic Biopsies of the Prostate. Cancers 2021, 13, 5685. https://doi.org/10.3390/cancers13225685. Strangely this work is not cited in the MS. 

Other similar studies that could have been cited are:

https://doi.org/10.1016/j.euf.2020.08.013

doi: 10.21037/tau-20-1237

https://doi.org/10.1007/s00428-019-02738-y
